# Prediction of Work-Related Risk Factors among Bus Drivers Using Machine Learning

**DOI:** 10.3390/ijerph192215179

**Published:** 2022-11-17

**Authors:** Pradeep Kumar Hanumegowda, Sakthivel Gnanasekaran

**Affiliations:** 1School of Mechanical Engineering, Vellore Institute of Technology, Chennai 600127, India; 2Centre for Automation, School of Mechanical Engineering, Vellore Institute of Technology, Chennai 600127, India

**Keywords:** decision tree, random forest, naïve Bayes, BMTC, machine learning

## Abstract

A recent development in ergonomics research is using machine learning techniques for risk assessment and injury prevention. Bus drivers are more likely than other workers to suffer musculoskeletal diseases because of the nature of their jobs and their working conditions (WMSDs). The basic idea of this study is to forecast important work-related risk variables linked to WMSDs in bus drivers using machine learning approaches. A total of 400 full-time male bus drivers from the east and west zone depots of Bengaluru Metropolitan Transport Corporation (BMTC), which is based in Bengaluru, south India, took part in this study. In total, 92.5% of participants responded to the questionnaire. The Modified Nordic Musculoskeletal Questionnaire was used to gather data on symptoms of WMSD during the past 12 months (MNMQ). Machine learning techniques including decision tree, random forest, and naïve Bayes were used to forecast the important risk factors related to WMSDs. It was discovered that WMSDs and work-related characteristics were statistically significant. In total, 66.75% of subjects reported having WMSDs. Various classifiers were used to derive the simulation results for the frequency of pain in the musculoskeletal systems throughout the last 12 months with the important risk variables. With 100% accuracy, decision tree and random forest algorithms produce the same results. Naïve Bayes yields 93.28% accuracy. In this study, through a questionnaire survey and data analysis, several health and work-related risk factors were identified among the bus drivers. Risk factors such as involvement in physical activities, frequent posture change, exposure to vibration, egress ingress, on-duty breaks, and seat adaptability issues have the highest influence on the frequency of pain due to WMSDs among bus drivers. From this study, it is recommended that drivers get involved in physical activities, adopt a healthy lifestyle, and maintain proper posture while driving. For any transport organization/company, it is recommended to design driver cabins ergonomically to mitigate the WMSDs among bus drivers.

## 1. Introduction

Conditions affecting the body parts responsible for movement—the muscles, tendons, bones, joints, ligaments, and nerves—are referred to as musculoskeletal diseases. Conditions affecting the musculoskeletal system can range from mild inconveniences to permanent disabilities. WMSDs are a leading reason for lost work time. The monetary burden of musculoskeletal illnesses on the public health care system is substantial. Musculoskeletal problems can affect any body part, although they often connect to the individual’s line of work. Vibration, for instance, has been linked to lower back diseases caused by repetitive motions such as lifting and carrying heavy objects. Repetitive or prolonged static force exertion can cause or exacerbate upper limb problems (the fingers, hands, wrists, arms, elbows, shoulders, and neck). These conditions range in intensity from mild, episodic annoyances to serious, clinically identifiable illnesses. When pain strikes, it might be the consequence of an acute overload that can be easily reversed, or it could indicate the onset of a more severe condition [1].

Health issues arise, in particular, when the mechanical workload exceeds the capacity of the musculoskeletal system’s components to absorb that stress. Expected outcomes include fractures, undetected microfractures, degenerative changes, and injuries to the bones, ligaments, and muscles (e.g., strains, ruptures). Additionally, functional limitations and early bone and cartilage degradation (including that of the menisci, vertebrae, intervertebral discs, and articulations) may occur. These irritations can develop at the places where muscles, tendons, and tendon sheaths enter. The two fundamental injuries are acute/painful and the other is chronic/lingering. The first kind is brought on by a powerful, brief heavy load that causes an abrupt breakdown in structure and function (for example, ripping a muscle owing to a hard lift, breaking a bone due to a fall, or immobilizing a vertebral joint due to a vigorous movement). The second is brought on by a persistent overload, which leads to discomfort and dysfunction that keeps getting worse (e.g., wear and tear of ligaments, muscle spasm, and hardening). Long-term loading can generate chronic injuries that the worker may reject or neglect because the damage may appear to heal fast and not cause a substantial disability [2].

Due to technological improvements that have increased the volume of data being collected and enhanced processing power, machine learning (ML) is a crucial subfield of artificial intelligence that is seeing significant usage in several sectors. A performance criterion can be optimized using methods in machine learning (ML) by using training data and/or prior knowledge [3]. ML techniques use computer programs to “learn” from current data using hyper-parameters chosen by the researchers, then they build models that either reveal the underlying structure (unsupervised learning) or predict discrete or continuous output variable(s) in unseen data (supervised learning). Common applications for machine learning algorithms include regression, classification, clustering, and reinforcement learning. By utilizing k-means clustering to identify subgroups, for instance, unsupervised learning may be able to identify hidden patterns in data sets [4].

Since ML can simulate non-linear interactions between a range of factors, it is thought that it is more suited for understanding the complex etiology of work-related musculoskeletal disorders (WMSDs), therefore preventing their recurrence [5]. Although interpretability for prediction performance can occasionally be compromised by ML approaches [6], these methods have already advanced the first WMSD preventative measures [7]. Due to its usefulness in addressing research concerns unrelated to the prevention of WMSDs, ML approaches have the potential to aid in WMSD prevention. Musculoskeletal problems accounted for 30% of DAFW (days absent from work) occurrences in several industrialized nations [8]. For many workers in many industry sectors, workplace injuries, occupational health issues, and other risk factors at work might cause pain and discomfort in the arms, shoulders, neck, back, or other essential organs [9].

Professional drivers are more vulnerable to the effects of weariness, which contributes to a greater rate of commercial vehicle collisions [10]. Bus drivers have sensitive and demanding jobs that are complicated by a variety of risk factors, including potential traffic jams, a lack of knowledge about transportation schedules, irregular work schedules, complicated routes, inclement weather, and an increase in the density, intensity, and speed of the flow of traffic. The primary physical and occupational risk factors were noise, vibration, toxic chemicals in the driver’s cabin, and driver anxiety [11]. Across the world, musculoskeletal problems are prevalent among the working population. The most important thing for every worker or operator to do on the job is to maintain appropriate posture [12]. Uncomfortable postures were found to be the main problem that employees and operators experienced at work [13]. In many quickly growing nations where labor is inexpensive, occupational health and safety norms and posture analysis tools are frequently disregarded [14,15].

Long periods of sitting and exposure to vibrations of various intensities are associated with the development of work-related musculoskeletal diseases in drivers [16]. Due to the nature of their jobs, bus drivers frequently have lower and upper extremity symptoms of musculoskeletal diseases [17]. Long-distance drivers typically have greater rates of work-related musculoskeletal diseases, particularly low back discomfort [18,19,20]. The likelihood of developing musculoskeletal problems in professional drivers can be directly influenced by ergonomic factors in vehicle design, individual driving behavior, and road conditions [21].

Workplace musculoskeletal problems may have a significant negative impact on communities, employers, and employees [22]. Using experimental and statistical techniques, prior research has demonstrated the link between workplace risk factors and the frequency of musculoskeletal illnesses among bus drivers [23,24,25,26]. There are a lot of opportunities to use modern technologies in ergonomic research nowadays because artificial intelligence and information technology are developing so quickly and because the difficulties and possibilities in the research environment are always changing [27].

There is a paucity of research on transport ergonomics and the incidence of WMSD in this sector, much of which focuses on specific risk factors such as stress, low back pain, etc. Second, it would seem that no studies have been conducted on the prevalence of WMSDs among BMTC bus drivers who use various machine algorithms. Finally, there are limited but erratic findings on the factors that influence WMSD among bus operators. It was hypothesized that:In bus drivers, work-related factors appear to be significant determinants of WMSDs.Model to predict the risk factors contributing to WMSDs among bus drivers.

In previous studies using various machine learning techniques, some of the ergonomic risk factors of workers and operators were evaluated, assessed, and analyzed by sensing-based activity assessment, motion analysis, mental load evaluation, risk stratification of physical workload, and fatigue classification methods [28].

The aim of this study was to identify the risk variables at work that may contribute to the occurrence of WMSDs in bus drivers. The main work-related risk variables were identified using a variety of machine learning approaches, including decision trees, random forest, and naïve Bayes algorithms. The results of the current study will be useful for identifying risk factors associated with the workplace and for conducting future experimental research to evaluate the severity of the risk factors.

## 2. Methods

This section consists of a data set, which was the primary asset for this study. From the data set, feature extraction was performed to select the appropriate features, and in subsequent stages, data pruning was performed as per the standard procedure. The brief overview is expanded on in the following sections.

### 2.1. Data Set

This study used a random sampling technique to determine its sample size (*n* = 400) [29]. There are now 5175 drivers working for BMTC between the east and west sides of Bengaluru in south India. The survey questionnaire was distributed to all 5558 full-time BMTC bus drivers in the east and west zones, who are between the ages of 24 and 55 years. It was standard practice to schedule drivers for 9–10 h each day, six days on and one day off. Drivers with asthma, diabetes, arthritis, or high blood pressure, those undergoing treatment for multiple conditions (piles or skin allergy), and those over the age of 55 were all ineligible. Due to inappropriate responses, 30 samples were discarded.

In total, 370 male bus drivers were studied, and it was discovered that 10% of them were between the ages of 24 and 28, 50% were between the ages of 29 and 39, and 39% were over 40. The participants’ mean (SD) height, weight, and age were 1.70 (SD +0.4) meters, 69.4 (SD +7.8) kg, and 40 (SD +6.8 years), respectively.

Permission from BMTC was received by the researchers of this study. All BMTC drivers were contacted by the Head of Human Resources and invited to participate in the survey. The study’s objectives were clearly explained to all participants, and their participation was entirely voluntary and anonymous. The questionnaire was examined by the Chief Engineer, the Head of Human Resources, the Depot Managers, and the Divisional Controller (DC) of BMTC. In total, 10% of the research sample was questioned one-on-one utilizing the questionnaire during their leisure time after obtaining the relevant clearances from BMTC. These people were a part of the larger research as well. This research was undertaken after the questionnaire was revised based on comments from the drivers and data collected in the field. Drivers’ raw data were partitioned so that only aggregate statistics would be used; individual identifiers would be removed at a later time.

### 2.2. Feature Extraction from the Data

Data for the study were gathered through face-to-face interviews, questionnaires, and direct observations. The Modified Nordic Musculoskeletal Questionnaire (MNMQ) served as the basis for the questions [30,31]. While examining the related risk variables of driving, drivers’ perspectives, experiences, and prior occurrences were taken into account. Three components make up the questionnaire: (a) demographic and socioeconomic data; (b) information about occupation, behavior, and lifestyle; (c) information about medical/health (history).

### 2.3. Machine Learning Techniques for Risk Factor Prediction

Artificial intelligence (AI) is a broad term that includes a subset known as machine learning. It gives computers and computing systems the capacity to learn and grow on their own without being explicitly programmed by humans. Machine learning is concerned with the use of data-driven methodologies to develop autonomous systems that can assist humans in making decisions with or without human supervision. Machine learning employs a set of algorithms and approaches to uncover and establish repeating patterns in data to create these autonomous systems. The supervised machine learning strategy is one of the most common and powerful machine learning approaches. An algorithm is given a collection of inputs called features/attributes and their associated outputs called target variables in supervised machine learning. A supervised machine learning approach is used to train a model that captures the complicated connection between the characteristics and target variables represented by a mathematical formula using a given data set. This trained model serves as the foundation for prediction. Predictions are created by using the trained model to generate the target variable from an unknown collection of features.

#### 2.3.1. Decision Tree

Although there are many other kinds of supervised classifiers, one of the most well-known is the decision tree (DT), which was developed by Quinlan. Judgmental trees are built by the DT using training data using the “entropy drop” or “info gain” concept. The term “training set” refers to a collection of samples for which the classification has already been made (i.e., the fault type is known). To illustrate, let us pretend that (A = a_1_, a_2_, ..., a_n_) is a training data set, and that each sector a_i_ is a k-dimensional vector (b_1,i_, b_2,i_, b_3,i_, ..., b_k,i_). Each b_j_ also indicates the category that a_i_ falls under, so together they reflect the attribute values of a certain industry. The decision tree seeks to partition a pool of data points into many groups, one group for each of the nodes in the network. In other words, there is a class to which each subset belongs (in our case, classes are faults). The entropy difference between the two systems is used as the dividing line. For this case, we used the criterion where the entropy difference was the greatest. If the classifier fails to successfully place all instances of a node into a single class, the procedure is restarted with a different node in mind. Here are a few of the algorithm’s most notable characteristics:The classifier builds a leaf node by offering obvious categorization if all of the sectors in the combination belong to the same class.If none of the characteristics add any new information, the algorithm makes a “decisive node”.If the difference in entropy between all characteristics is zero, the algorithm creates a decisive node based on the predicted identity of the class.

#### 2.3.2. Random Forest

This algorithm makes connections between different decision trees and then combines them to make more accurate, trustworthy, and reliable predictions. It employs both classification and regression; therefore, it is viewed as a combined approach. The model’s training process becomes robust as a result of the extra randomness that is added as a tree-like pattern develops. The key characteristic is that, rather than dividing the node, it finds the most noticeable feature between random subdivisions. This wide-ranging selection creates the finest model. Here, the initial phase is the simultaneous development of many data set subdivisions and accompanying decision trees. It goes without saying that there will be n decision trees for n sub-divisions. Finding the mean of all the data is the final step. It combines the straightforward idea of trees with their flexible nature, improving accuracy. It functions as a massive array of de-correlated random trees. It generates a large number of random trees and decides based on the mode of the classes. Overfitting is an issue that is resolved by random forest, which can also handle huge data sets with higher dimensionality. Because it outperforms other algorithms in terms of accuracy, the random forest is particularly well liked. It works well with bigger data sets and can evaluate different input qualities without any compromise in performance. Additionally, it determines the crucial characteristics for accurate categorization. It calculates an internal unbiased for generalization error as the forest structure expands. It maintains the model’s accuracy when a higher fraction of the data is absent. This is accomplished by approximating the missing data. Additionally, it makes balancing the mistakes in imbalanced sets easier. It determines set relationships that may be used for scaling, tracing outliers, and assembly. The random forest technique may be extended to analyze unlabeled data sets to provide unsupervised learning.

#### 2.3.3. Naïve Bayes Classifier

Bayes classifiers are classification algorithms that employ the Bayes theorem when there is a high degree of independence between the data components. These classifiers comprise the probabilistic classifier family. A Bayesian classifier generates a probabilistic model by creating links between the features. If these correlations are understood beforehand, creating the model becomes significantly simpler. The generated model is then used to predict the classification of newly fed examples. The naïve Bayes classifier is a variant of this family in which the data instances are conditionally independent and have no hidden features that influence their categorization. The characteristics are also discrete variables. The fundamental concept underlying the model is to determine the likelihood of an instance belonging to a given class. Using Bayes’ theorem, this probability is computed. When a problem has more than one class, the instance is given the name of the class with the highest probability.

### 2.4. Feature Selection

The method of feature selection is crucial to machine learning. The feature selection procedure may be used to increase the accuracy ratings of estimators (classifiers) or to enhance their performance on data sets with extremely high dimensionality. It is beneficial to use all of the input features, including those that are unimportant, when there is enough data and time to do so, to approximate the principal function between the input and the output. The irrelevant characteristics present two issues:The cost of calculation will increase.The training procedure could be misled by the irrelevant input characteristics.

Therefore, to keep the size of the approximator model minimal, those input characteristics with little impact on the output may be disregarded. Therefore, choosing the right features is crucial in determining how accurate a classification will be. As a foundation for the investigation, a reasonably diverse set of statistical factors was chosen. Mean, standard deviation, median, standard error, variance, kurtosis, skewness, range, minimum, maximum, sum, and count are some of the terminologies used to characterize them. This information came from the analysis of questionnaire data. There is no need for every feature to show all of the important information. In general, certain traits could reveal more data than others. Feature selection is the process of choosing such superior characteristics that expose more data for categorization. Because each feature contributes a dimension to the feature space, and only choosing a small number of features minimizes the dimension, this approach is also known as “dimensionality reduction”.

### 2.5. Pruning

In the original data set, 30 characteristics were collected from respondents’ personal and professional lives. The data has been cleansed using industry-standard procedures to ensure the reliability of the replies [32]. To narrow down to a workable model, we had to disregard some of these characteristics. In the end, we settled on 21 of the aforementioned features. To quantify the relative importance of the various textual replies, weights were assigned. ‘Yes’ is counted as 1, and ‘No’ as 0. A label encoder was used to transform the categorized information into numerical form. Finally, 70% of the data was used to train the model, while the other 30% was used for testing.

## 3. Data Analysis

The responses from the drivers were coded once they were entered into the survey form and used for the proper analysis. The initial stage of data analysis includes evaluating the risk factors contributing to the prevalence of WMSDs among bus drivers to test the first hypothesis of the study. This study used SPSS v.23 to conduct a chi-square (χ^2^) test (with a 95% confidence interval) to investigate the relationship between the independent and dependent variables. A chi-square test has been conducted and risk factors with *p*^a^ < 0.05 were considered significant risk factors. A total of 21 features/risk factors were identified which were derived from the first hypothesis. Several independent variables were included in this study: sociodemographic, workplace, and lifestyle/behavior/occupational data. However, reported musculoskeletal problems were taken into account as an independent variable. *p* 0.05 was judged to be statistically significant in the χ^2^ test findings, which is shown in the Appendix A.

## 4. Result

The BMTC bus drivers’ data were first analyzed to identify risk indicators for WMSDs. The prevalence of WMSDs from the previous 7 days was also gathered during the questionnaire survey. Predictions for the first hypothesis testing were confirmed through the chi-square test. To identify the most significant risk factors, commonly known machine learning algorithms were used, and the confusion matrix of respective algorithms shows the features categorization accuracy. The second hypothesis was tested using machine learning algorithms. In the below-mentioned sections, there is a brief description of the results found at each stage of data analysis.

### 4.1. Data Extracted from the Questionnaire

In total, 22% of the drivers were judged to be in good health, while the remaining drivers had ordinary health status with a history of WMSDs. In their spare time, 33% of drivers participated in outdoor sports.

For more than half of the study participants, their daily jobs resulted in a loss of strength and muscular fatigue when driving. Drivers reported sleeping on bus floors (passenger compartment) at the end of their shift in part because of unhygienic rest stops at depots and their own residences that were far from the place of employment; 0.78% of drivers reported turning off the ignition at traffic signals. More than a fifth of drivers (21%) said they drove erratically to complete their scheduled trips, and more than half (57%) said the heat from the engine, the lack of cabin ventilation, and the overall weather conditions caused them to feel uncomfortable inside the driver cabin. The present survey found that 73% of drivers sat for prolonged periods, 77% had seat adaptation (adjustment) concerns, 71% were not happy with their employment, 85% were under stress because of their busy schedules and the traffic, and 87% relied on outside meals. Additionally, 72% said they did not have enough access to restrooms and drinking water while at work, 81% said they were subjected to vibration from the vehicles and the roads, and 73% said they had trouble getting in and out of buses.

#### Prevalence of WMSDs from the Previous 7 Days

In order to obtain self-reported objective evaluations of pain and discomfort, the body part discomfort (BPD) scale was employed [33,34]. Drivers are deemed to have WMSDs if their symptoms are rated 1–5 (1—Not unpleasant, 2—Barely uncomfortable, 3—Quite painful, 4—Very uncomfortable, and 5—Extremely uncomfortable). It was discovered that drivers had experienced symptoms in their shoulder (23%), neck (13%), arm (9%), forearm (8%), upper back (15%), hip/buttocks (25%), lower back (32%), thighs (17%), knees (19%), fingers (9%), and ankle/foot (9%) within the preceding seven days. Additionally, it was shown that individuals had the fewest symptoms (6%) in the hand/wrist region.

### 4.2. Classification Using Decision Tree

Figure 1 depicts the trained decision tree structure. The number of nodes in the tree is 21, and the number of leaves is 11. The training begins with a test at the root node with the feature “participation in physical activities”. The test determines if “Involvement in physical activities is less/greater than or equal to 0”. If the result is “less than 0”, the next test is performed on “tobacco consumption” for the following values. Similarly, tests are performed at all nodes. The training time was 0.03 s.

Based on ten-fold cross-validation, the correctly categorized occurrences are 357/357, i.e., 100%, with no wrongly identified instances, and the classification results are detailed in Table 1. The confusion matrix for the decision tree is shown in Table 2. The decision tree design in this training suggests just nine features, which are as follows:Involvement in physical activities.Tobacco consumption.Frequent posture change.Egress/ingress.Exposure to vibration.On duty breaks.Seat adaptability issues.Tired at end of the work.Sleeping in the bus (after duty).

### 4.3. Classification Using Random Forest

The confusion matrix produced by the random forest tree classifier with 10-fold cross-validation is shown in Table 3 and Table 4 and is self-explanatory. The classification accuracy is 100% since 357/357 cases were properly classified.

### 4.4. Classification Using Random Forest

As shown in Table 5 and Table 6, the confusion matrix was created using the naïve Bayes classifier with 10-fold cross-validation.

For the class “VERY OFTEN”, all 50 instances are appropriately categorized; thus, there is no misclassification. For the condition “OFTEN”, 104 instances are correctly classified and 14 are misclassified as “SOMETIMES”. For the class “SOMETIMES”, only 78 instances are correctly classified, whereas 7 were misclassified as “RARELY”. Further, for the class “RARELY”, only 27 instances are correctly classified, whereas 3 were misclassified as “SOMETIMES”. For the class “NEVER”, 78 instances are correctly classified.

### 4.5. Comparative Analysis

Machine learning techniques discussed in the earlier section were tested in the Weka tool to predict significant work-related physical risk factors on the frequency of pain in the last 12 months. In the trained model, frequency of pain was considered as target variable, and significant work-related physical risk factors were considered as attributes. The response from the drivers for frequency of pain was coded as follows: Very often—1, Often—2, Sometime—3, Rarely—4, and Never—5. It can be inferred that all the trained models performed fairly well in classification, among which decision tree (Figure 1) and random forest techniques achieved 100% accuracy followed by naïve Bayes with 93.28% accuracy. Results from the first hypothesis testing were utilized as inputs for the second hypothesis. To identify the feature importance, 21 attributes were considered, and results from the decision tree show that risk factors such as involvement in physical activities, frequent posture change, exposure to vibration, egress ingress, on-duty breaks, and seat adaptability issues have the highest influence on the frequency of pain due to WMSDs among bus drivers.

The above work-related risk factors show the highest significance to the prevalence of WMSDs among bus drivers. Hence, the second hypothesis yields a positive result.

Based on the above comparative analysis, some algorithms yield overfitting and some underfitting in the model. Since authors have considered all 21 features and the frequency of prevalence of WMSDs in the past 12 months as target variables for all the drivers’ responses without modifying them, any stage may be a possible reason for overfitting or the high performance of algorithms.

All drivers’ replies were unfiltered in the final study to prevent data tampering. Initial training of algorithms was performed on the training data, and both the decision tree and the random forest produced the same outcome despite their contrasting strengths of low bias and high variance. Machine learning and data science have one main flaw, overfitting. Overfitting on the training data sets is a typical issue when using decision trees for classification and regression problems since they are a non-parametric supervised implementation approach. Given the model’s architecture, if the model is allowed to be trained to its full power, the model is practically guaranteed to overfit the training data. Fortunately, overfitting in machine learning algorithms may be avoided and prevented using a number of different methods [35,36,37,38,39,40,41,42,43,44]. Some methods that are frequently employed to prevent overfitting in decision trees are as follows:Pre-pruning.Post-pruning.Acquire more training data.Remove irrelevant attributes.Cross validation.

1.Pre-Pruning

This method prevents the non-important branches from growing. According to the specified condition, it ends the formation of new branches. In total, 92% classification accuracy was obtained by this process, which fixes the overfitting of the decision tree model.

2.Post-Pruning

The non-significant branches are first trimmed or eliminated once the whole tree has been formed. Cross validation is performed at each stage to see whether the new branch’s inclusion increases accuracy. Classification accuracy of 91% was achieved using this method.

3.Removal of features

In the present study, 21 features were used as input attributes for machine learning algorithms. No significant changes were observed when attributes were reduced to 18. After ≤18, the classification accuracy was significantly changed in all the algorithms used in the present study. The classification accuracy was 92% for the decision tree algorithms.

4.Increasing the trained data set

In the present study, 70% of data is considered a trained set; when the same is increased to 80%, the classification accuracy decreased to 96% for the decision tree algorithm.

5.Stratified cross validation

Twenty characteristics generated from the chi-square test were used in a stratified K-fold cross validation test performed by the authors. Twenty characteristics were divided into five four-characteristic groupings. Each data set at each testing stage has been designated as the validation set, while the remaining data sets have been designated as the test sets, each data set has an equal opportunity to serve as a validation set in future testing. In a 5-fold cross-validation test, accuracy was determined to be around 96% on average. For the class “VERY OFTEN”, all 50 instances are appropriately categorized; thus, there is no misclassification. For the condition “OFTEN”, 104 instances are correctly classified and 15 are misclassified as “SOMETIMES”. For the class “SOMETIMES”, only 84 instances are correctly classified. Further, for the class “RARELY”, 30 instances are correctly classified, and for the class “NEVER”, 74 instances are correctly classified. The results are depicted in Table 7 and Table 8. Since there was a very small number of traits to begin with (just 20), it made sense to divide them into five groups (K = 5). The references are included below for your convenience.

The summary of the results from all the above techniques to prevent overfitting is given in Table 9. Similarly, in this study, random forest yields 100% classification accuracy, indicating that overfitting in the model can be addressed by the below methods, and the results are depicted in Table 10.

Reduce tree depth.Reduce number of variables sampled in each split.Acquire more training set.K-fold cross validation test.

#### Independent Variables Validation

The independent variables in this study were selected from the variables that have significant association with WMSDs among BMTC bus drivers. Initially, the 70% trained data set and 30% test data set was used to check the classification accuracy. Decision tree and random forest algorithm yields were overfitting, so to validate the independent variables, percentage of the trained and test data set were changed and a few iterations were conducted using decision tree and random forest algorithms; the results are given below in Table 11.

## 5. Discussion

According to the study’s findings, 67% of drivers had WMSD in the previous 12 months. The results are better than what was noted in African nations. In Nigeria, Akinpelu et al. observed a frequency of 64.8% [45], while in Ghana, Abledu et al. recorded a prevalence of 59% [46]. Our number is larger than that which has been reported in Asia. For instance, Tamrin et al. found a frequency of 60.4% in Malaysia [47], while Jadhav et al. in India reported a prevalence of 67.4% among public bus drivers [48]. According to research by Grace Szeto et al. on WMSDs in urban bus drivers in Hong Kong, the neck, lower and upper back, knees and thighs, and shoulder have the highest incidence rates, ranging from 35 to 60%, and 90% of the discomfort were caused by bus driving [49]. In Pondicherry, 22.9% (*n* = 667) of drivers reported having low back discomfort [50]. Data analysis derived some dependent variables such as involvement in physical activities (67.02%), frequent posture change (74.59%), exposed to vibration (81.35%), egress ingress (73.5%), on-duty breaks (48.92%), and seat adaptability issues (77.29%). In total, 63.78% of drivers reported that their muscles get fatigued during working hours due to continuous driving. Professional drivers will have the highest responsibility on the road because drivers should transport passengers safely and perform safe driving activities, by following the regulations of the traffic in congested roads within the city limits. These drivers are the most exposed to fatigue on a daily basis. Fatigue may be due to work factors, sleep factors, and health factors [51]. It was found that almost all drivers worked more than 50 h per week. On average, drivers spent 10–12 h behind the wheel each day. Indian labor laws mandated that average working hours for 4 consecutive months must not be longer than 48 h/per week. BMTC has a target of 200 km/day/bus assigned for each driver. However, due to metro rail construction, heavy traffic, and congested roads with potholes, the assigned target cannot be reached within working hours. To compensate for the loss, drivers must extend their shifts. The result from the data analysis and machine learning technique yields that involvement in physical activity is one of the predominant work-related risk factors affecting WMSDs. It was found that 67.02% of drivers have no habit of doing regular exercises daily. Many studies have proved that regular exercise will decrease musculoskeletal discomfort [52,53]. Only ergonomic interventions were not sufficient to eliminate/mitigate the WMSDs among bus drivers, but along with organizational factors and individual factors, chances are high to reduce musculoskeletal discomfort [54]. The World Health Organization (WHO) recommends that people are involved in physical activity at latest 150 min/per week [55,56]. A lack of physical activity may lead to hypertension, stress, and diabetes followed by cardiovascular diseases [57,58]. In total, 73.5% of drivers reported that they have been exposed to vibration on a daily basis. Long-term exposure to whole-body vibration leads to the development of WMSDs among bus drivers. Experimental studies have reported the relation between the speed of the bus, type of roads, and vibration magnitude of their influence on WMSDs on bus drivers [59]. Several studies proved the relationship between the types of seats and whole-body vibration exposure. Results show that there are no ideal seats that suit various kinds of passenger vehicles [60]. An electromagnetically active (E-active) seat, which is a relatively new design in the industry, has a promising feature to dampen vibration significantly compared to other kinds of seats [61]. All recent studies on whole body vibration show that higher vibration magnitude was recorded in the vertical axis (+*Z*-axis) direction than in lateral and fore-aft directions [62]. Ergonomic interventions and engineering controls in designing the seats and vibration dampers can resolve the issue of vibration exposure among bus drivers.

The intersection of machine learning and biomechanics offers great promise for ergonomics research and accelerates rehabilitation programs for musculoskeletal diseases. In the present study, a 70% training data set and 30% test data set is considered, and in several studies, an additional validation data set was used to select features or hyperparameters [63]. Machine learning techniques/methods were used ranging from passively monitoring post-stroke patients to predicting WMSDs among the different working populations [64]. The research aimed to highlight machine learning efforts in predicting the physical risk factors contributing to WMSDs among bus drivers. In the present study, decision tree, random forests, and naïve Bayes algorithms were used to predict the work-related risk factors contributing to the WMSDs. These algorithms were commonly used in recent studies related to occupational health and safety. There are many other algorithms such as ANN (artificial neural network), SVM (support vector machine), CNN (convolutional neural network), etc. Using smart technology such as machine learning, it is possible to predict the potential risk at workplaces and also reduce cumulative exposure of workers to occupational and health safety risk. In the present study, a field survey was conducted through face-to-face interviews with bus drivers at BMTC to gather data on risk factors. The responses were based on the experience of drivers and a better understanding of the purpose of the study. In a larger vision to extend the present study, it is possible to evaluate the actual risk at the workplace by using wearable sensors, SEMG (surface electromyography) to evaluate muscle activation, joint movements, and muscle fatigue. In the present study, authors have limited the scope of the study to collecting the data from drivers and analyzing the data to validate the response from data analysis and simple machine learning techniques.

## 6. Limitations

The current study relied solely on drivers’ self-reported responses; no clinical data were collected.Since most of the driver’s replies to the physical risk factors were binary, future research on this topic may include gathering data based on the frequency, seriousness, and intensity of the risk factors.The study’s participants were all male drivers.

## 7. Conclusions and Recommendation

The primary concern of this study was to identify work-related risk variables linked to WMSDs in bus drivers using machine learning approaches. It was found that 66.7% of BMTC drivers self-reported that they were suffering from WMSDs. The machine learning algorithms used in this study show that physical activities, frequent posture change, exposure to vibration, egress ingress, on-duty breaks, and seat adaptability issues have the highest influence on the frequency of pain due to WMSDs among bus drivers.

BMTC drivers can reduce their risk of WMSDs by adopting several preventative measures, including maintaining a healthy, relaxed posture while working, engaging in regular physical activity, participating in ergonomic training, and taking frequent breaks. Drivers must be urged to see physiotherapists for any musculoskeletal problems they are experiencing on the job.

## 8. Future Scope of the Study

Data were collected using self-reported questionnaires; there may be chances of overestimation in reporting the WMSDs.

(a)To verify the effectiveness of the model, other techniques such as KNN (K-closest neighbors), SVM (support vector machine), logistic regression, and ensembled techniques such as boosting and bagging classifiers can be utilized.(b)To assess the effectiveness of the model and confirm how the model works for the provided data set, deep learning techniques such as CNN may be employed.(c)Since the number of replies in our situation is constrained, a vast data set can be employed as a training model.

## Figures and Tables

**Figure 1 ijerph-19-15179-f001:**
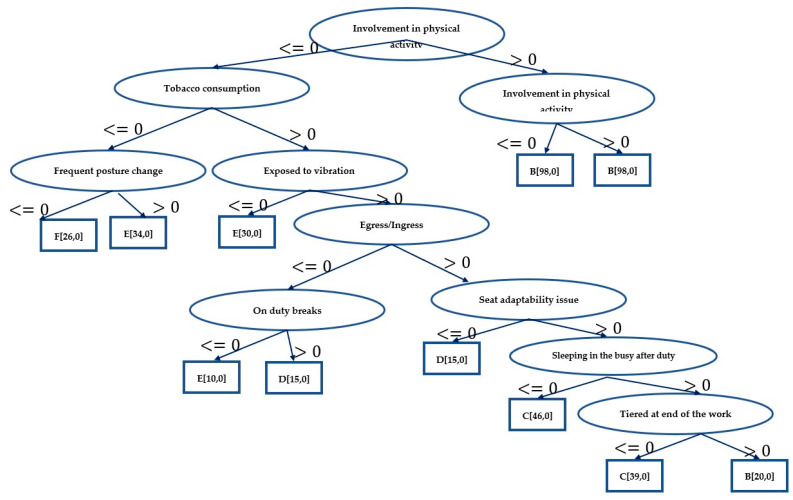
Decision tree model.

**Table 1 ijerph-19-15179-t001:** Classification result for decision tree.

** *Classification* **	
Number of samples trained	357
Accurately classified samples	357
Classification accuracy	100%
Wrongly classified samples	0
Misclassification	0
Inter-rater agreement using Cohen’s kappa	1
** *Errors* **	
Root relative square (RRSE)	0
Relative absolute (RAE)	0
Mean absolute (MAE)	0
Root mean square (RMSE)	0

**Table 2 ijerph-19-15179-t002:** Confusion matrix for decision tree.

Class	Very Often	Often	Sometimes	Rarely	Never
Very Often	50	0	0	0	0
Often	0	118	0	0	0
Sometimes	0	0	85	0	0
Rarely	0	0	0	30	0
Never	0	0	0	0	74

**Table 3 ijerph-19-15179-t003:** Classification result for random forest.

** *Classification* **	
Number of samples trained	357
Accurately classified samples	357
Classification accuracy	100%
Wrongly classified samples	0
Misclassification	0
Inter-rater agreement using Cohen’s kappa	1
** *Errors* **	
Root relative square (RRSE)	1.90%
Relative absolute (RAE)	0.11%
Mean absolute (MAE)	0.0003
Root mean square (RMSE)	0.0068

**Table 4 ijerph-19-15179-t004:** Confusion matrix for random forest.

Class	Very Often	Often	Sometimes	Rarely	Never
Very Often	50	0	0	0	0
Often	0	118	0	0	0
Sometimes	0	0	85	0	0
Rarely	0	0	0	30	0
Never	0	0	0	0	74

**Table 5 ijerph-19-15179-t005:** Classification result for naïve Bayes.

** *Classification* **	
Number of samples trained	357
Accurately classified samples	333
Classification accuracy	93.28%
Wrongly classified samples	24
Misclassification	6.72%
Inter-rater agreement using Cohen’s kappa	0.9138
** *Errors* **	
Root relative square (RRSE)	0.1371
Relative absolute (RAE)	12.12%
Mean absolute (MAE)	0.0313
Root mean square (RMSE)	38.16%

**Table 6 ijerph-19-15179-t006:** Confusion matrix for naïve Bayes.

Class	Very Often	Often	Sometimes	Rarely	Never
Very Often	50	0	0	0	0
Often	0	104	14	0	0
Sometimes	0	0	78	7	0
Rarely	0	0	3	27	0
Never	0	0	0	0	74

**Table 7 ijerph-19-15179-t007:** Classification result of stratified cross validation.

** *Classification* **	
Number of samples trained	357
Accurately classified samples	342
Classification accuracy	95.79%
Wrongly classified samples	15
Misclassification	4.21%
Inter-rater agreement using Cohen’s kappa	0.9118
** *Errors* **	
Root relative square (RRSE)	0.1261
Relative absolute (RAE)	10.12%
Mean absolute (MAE)	0.0313
Root mean square (RMSE)	35.16%

**Table 8 ijerph-19-15179-t008:** Confusion matrix of stratified cross validation.

Class	Very Often	Often	Sometimes	Rarely	Never
Very Often	50	0	0	0	0
Often	0	104	15	0	0
Sometimes	0	0	84	0	0
Rarely	0	0	0	30	0
Never	0	0	0	0	74

**Table 9 ijerph-19-15179-t009:** Results of preventive techniques (decision tree).

Techniques to Prevent Overfitting in Decision Tree	Accuracy %
Pre-Pruning	92
Post-Pruning	91
Acquire more training set	95
Remove irrelevant attributes	92
K-fold cross validation test	96

**Table 10 ijerph-19-15179-t010:** Results of preventive techniques (random forest).

Techniques to Prevent Overfitting in Random Forest	Accuracy %
Reduce tree depth	96
Reduce number of variables sampled in each split	91
Acquire more training set	94
K-fold cross validation test	93

**Table 11 ijerph-19-15179-t011:** Iteration on independent variables.

% of Trained Data— % Test Data	Accuracy %	
	Decision Tree	Random Forest
70–30	100	100
80–20	96	94
90–10	98	95
60–40	89	86

## Data Availability

Not applicable.

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
