# Peer review of "Prediction of Work-Related Risk Factors among Bus Drivers Using Machine Learning"

_ijerph, 2022, doi:10.3390/ijerph192215179_

Round 1

Reviewer 1 Report

The study investigating the work-related risk factors among bus drivers using machine learning is insightful. I have several comments below.

1. Could authors add more description of WMSDs with significant characteristics? What is the difference between yes and no? 

2. In the research design, the majority of features are binary such as stress at work, reliance on out-side food. These unmeasurable features may affect the generalization for risk factors. 

3. The authors trained different models using decision tree, random forest, naive bayes. It is suspicious that the accuracy reached 1 due to model overfitting. Could authors explain the high performance? 

Author Response

Dear Sir,

The authors would like to thank the journal's editor and reviewers for considering our work for publication in "Environmental Research and Public Health". The authors expressed their sincere thanks to the reviewers for finetuning this manuscript. We have taken utmost care to address all the comments raised by the reviewers with suitable justifications. The changes that we are made are highlighted in red color. We shall be happy if our revised manuscript is accepted for publication in view of our explanations in this reply and revision of the manuscript. The detailed information is given below for your kind reference.

Reviewer 2 Report

Dear authors, thank you for the opportunity to get acquainted with your work. The topic is undoubtedly relevant and in demand.

At the same time, there are a number of suggestions for changing the structure of your manuscript for better understanding.

The introduction contains information about the relevance and significance of the study for science and practice, which is described based on modern scientific research. It ends with the purpose of the study, at the same time, the authors do not indicate which hypotheses of the study, which makes it difficult to analyze the results.

Sections 2, 2.1., 2.2. and 2.3. will place below, after the authors state the idea of ​​the study itself, a sample. Currently sections 2, 2.1, 2.2. and 2.3. look isolated and do not fit into the overall logic of the presentation.

Section 3.1. The Data Set should be completed with a clearer description of the final sample - the total number and distribution of the sample by gender, age, driving experience, etc.

Section 4. Data analysis usually contains information about the statistical methods used and the logic of their application to test hypotheses, while the tables with values ​​themselves are described in the results section. At the same time, Table 1 is very voluminous and difficult to understand, it is proposed to change it, make it more concise and, as an option, mark it in Appendix 1 to the article, or include only a part in the article itself, and complete information in the Appendix.

Because Since authors have a separate Discussion section, the Result and Discussion section should be renamed to Results only. After the authors indicate the hypotheses of the study, it is necessary to revise the logic of the presentation of the results in accordance with the testing of hypotheses, and also add indications after testing each hypothesis, whether it was confirmed or not.

Each table or figure must be preceded by its description with a link to it; there should not be a row without a description of tables and figures.

Add study limitations and practical recommendations to the discussion of the results.

The conclusions should be supplemented in accordance with the results of the study.

Best regards, reviewer

Author Response

(The authors gave the same response as above.)

Round 2

Reviewer 1 Report

Thank you for the efforts from the authors making this study more clear. However, I have additional comments below. 

I previous commented that: "The authors trained different models using decision trees, random forest, naive Bayes. It is suspicious that the accuracy reached 1 due to model overfitting. Could authors explain the high performance?". In the response, the authors clarified about the feature selection and sample size increase processes, besides other approaches cross-validation, or employing ensemble approaches. 

1. Could authors provide the reference about the approaches to fix overfitting?

 2. The cross validation and independent data validation are usually more convincing after the performance comparison. Did authors have results from such approach?

3. Please add the responses to the main text properly.  

Author Response

Dear Sir,

Thanks for your  valuable feedback and hereby we attached the response with the revised manuscript

Reviewer 2 Report

Dear authors, thank you for your additions and corrections. All comments have been taken into account. The article is recommended for publication.

Best regards, reviewer

Author Response

Dear Sir,

Thanks to the reviewer for accepting our responses